# Spatio-Temporal Relationship between Land Cover and Land Surface Temperature in Urban Areas: A Case Study in Geneva and Paris

**Xu Ge, Dasaraden Mauree** \*,† 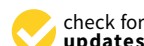, **Roberto Castello** and **Jean-Louis Scartezzini**

Solar Energy and Building Physics Laboratory, Ecole Polytechnique Fédérale de Lausanne,
CH-1015 Lausanne, Switzerland; s181411@student.dtu.dk (X.G.); roberto.castello@epfl.ch (R.C.);
jean-louis.scartezzini@epfl.ch (J.-L.S.)
\* Correspondence: dasaraden.mauree@gmail.com
† Current address: BG Ingénieurs Conseil, 1470 Lausanne, Switzerland.

**Abstract:** Currently, more than half of the world's population lives in cities, which leads to major changes in land use and land surface temperature (LST). The associated urban heat island (UHI) effects have multiple impacts on energy consumption and human health. A better understanding of how different land covers affect LST is necessary for mitigating adverse impacts, and supporting urban planning and public health management. This study explores a distance-based, a grid-based and a point-based analysis to investigate the influence of impervious surfaces, green area and waterbodies on LST, from large (distance and grid based analysis with 400 m grids) to smaller (point based analysis with 30 m grids) scale in the two mid-latitude cities of Paris and Geneva. The results at large scale confirm that the highest LST was observed in the city centers. A significantly positive correlation was observed between LST and impervious surface density. An anticorrelation between LST and green area density was observed in Paris. The spatial lag model was used to explore the spatial correlation among LST, NDBI, NDVI and MNDWI on a smaller scale. Inverse correlations between LST and NDVI and MNDWI, respectively, were observed. We conclude that waterbodies display the greatest mitigation on LST and UHI effects both on the large and smaller scale. Green areas play an important role in cooling effects on the smaller scale. An increase of evenly distributed green area and waterbodies in urban areas is suggested to lower LST and mitigate UHI effects.

**Keywords:** green urban infrastructure; satellite imagery; impervious surface; land surface temperature; urban heat islands; urban waterbodies

## 1. Introduction

The global urban population will increase from 4.2 billion in 2018 to 5 billion by 2030. More than half (55%) of the global population currently lives in urban areas. The ratio of urban-to-rural population is expected to increase to two-thirds (66.4%) by 2050, according to United Nations agencies [1]. Built-up areas host more than 50% of the population and 80% of the economic activity on earth. The global urban land cover is expected to increase by 1,527,000 km$^2$ by 2030 [2]. This will significantly impact the Land Surface Temperature (LST). LSTs are generally higher in urban areas compared to rural green area. This is due to the urban heat island effect, particularly during the summer time, leading to an impact on energy consumption, air quality and human health [3]. LSTs of waterbodies are generally lower. Consequently, a better understanding of the effects of the impervious surfaces and of green infrastructure on LST is necessary for mitigating adverse impacts and for supporting urban planning and public health management [4,5].

Remote sensing technologies are currently a core technology for Earth observation [6]. There are various remote sensing platforms and types of Earth observation sensors. They are used to obtain data on the global scale with high precision and high resolution, expanding the understanding of the environment from spatial and temporal perspectives [7]. Thermal remote sensing captures the radiation emitted from the ground in the 3–5 µm and 8–14 µm wavelengths [8], from which the land surface temperature can be determined. In addition, various indices, like the normalized difference built-up index (NDBI), the normalized difference vegetation index (NDVI) and the modified normalized difference water index (MNDWI), can be obtained from the remotely sensed images for monitoring the spatial distribution of land cover.

Landsat 5, Landsat 7 Enhanced Thematic Mapper Plus (ETM+) and Landsat 8 images have frequently been used in previous studies to derive LST. Kumar's study [9] used Landsat 5 and the Radiative Transfer Equation (RTE) method to derive LST without atmospheric corrections. Weng's study [10] characterized the annual and seasonal LST behaviors based on Landsat 5 images with less than 30% of cloud cover, using the RTE method with the NDVI Thresholds Method to obtain the emissivity values. Bokaie's study [11] used the Dark Object Subtraction method to correct satellite images atmospherically. LSTs were derived from Landsat 7 in Zhou's [12] and Zhang's studies [13] with different atmospheric correction methods. The authors of Yu's study [8] compared three different approaches for LST derivation using Landsat 8 satellite images, including the Radiative Transfer Equation-based method, the split-window algorithm and the single channel method. Empirical emissivity values were used in most of the studies. To avoid the atmospheric correction, clear sky images were frequently selected [11,12]. Landsat 8, carrying the newest thermal infrared sensor (TIRS), was rarely used in seasonal LST estimation and analysis of LST and land cover features.

Several studies have investigated the correlation between LST and land cover features. Estoque's study [14] applied urban-rural gradient analysis to determine the spatial variability of LST and the spatial distribution of impervious surfaces and green area across the urban-rural gradients. A multi-resolution grid-based analysis was used to examine the influence of impervious surfaces and green area on LST based on various sizes of polygon grids. Zhang's study [15] examines the spatial and temporal patterns of LST in the context of the urban heat island (UHI) phenomenon. The relationship of LST with six socio-ecological variables, i.e., land use/land cover, vegetation index, impervious surface index, water index, population density, and fossil-fuel $CO_2$ emission, were also investigated. A total of 5000 random points were generated across the study area in their study to explore the relationship of LST with NDVI, NDBI, and NDWI. A bivariate correlation analysis and scatter plots were used to determine the relationships between LST and variables. Due to various limitations, those studies focused on conducting analyses with 1-day land surface temperature data. There is however a lack of studies looking at how land cover features influence both the spatial and annual evolution of land surface temperature and urban heat islands effects. The present study explores seasonal effects of land cover features on both spatial distribution of LST and urban heat island effects in Paris and Geneva from a large to a smaller scale.

The specific objectives are to: (1) to derive the LST, NDBI, NDVI, and MNDWI, and analyze their spatial variations using Landsat 8 satellite images; (2) to apply a distance-based analysis to explore the temporal and spatial pattern of LST along transects in different study areas; (3) to apply a grid-based analysis to analyze the relationships between LST and the density of green area, and between LST and the density of impervious surfaces; (4) to apply the Spatial Lag Model (SLM) to investigate the spatial correlation of LST and dependent variables, i.e., NDBI, NDVI and MNDWI; and (5) to investigate the seasonal influence of various land cover features on LST and urban heat island effects. This study is expected to enhance the understanding of how LST varies with land cover features across different seasons and in two different settings.

## 2. Methodology

In this section, the region and relevant data sets used are introduced. The description of the variables definition, variables computation and the spatial analysis conducted at distance-level, grid-level and point-level are addressed.

### 2.1. Study Area

The focus of this study are the cities of Geneva and Paris (Figure 1). These two cities have similar climatic conditions and landscape compositions, such as large waterbodies within the city center. The area considered for Geneva (289 km$^2$) is more than 2.5 times larger than that of Paris (110 km$^2$), with a more complicated mixture of land cover features. It is worthwhile to compare how these affect the temporal-spatial pattern of LST in the two cities.

Geneva is located at the southwestern end of Lake Geneva (46°12′ N 6°9′ E), where the Rhône flows out. Its climate is temperate, classified as an oceanic one. Spring and winter are cool, summer is relatively warm, while autumn is slightly wetter than other seasons. There has not been any significant change in land use in Geneva from 2013 to 2018 [16], presented in Table 1. The total area of the studied region is around 289 km$^2$.

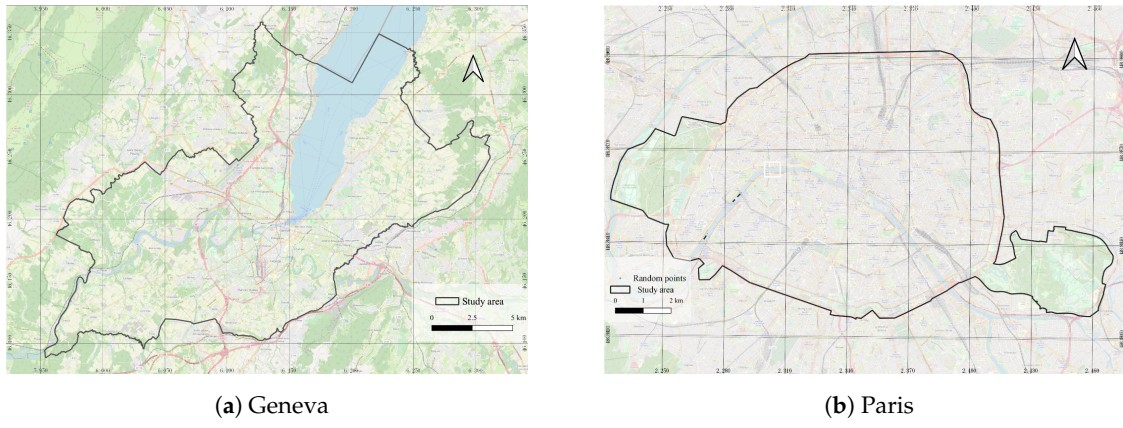

(**a**) Geneva          (**b**) Paris

**Figure 1.** Study area.

Paris is located in northern central France (48°51′24″ N 2°21′8″ E), in a north-bending arc of the river Seine. The city has a typical Western European oceanic climate, which is mild and moderately wet throughout the year [17]. Summer days are warm. Spring and autumn days are changing and unstable. Surprisingly, warm or cool weather occurs frequently. Winters are cool but the temperatures are on average above 0 °C [17]. The total area of the studied region is around 110 km$^2$. Land use statistics in 2017 are presented in Table 2. It was assumed that land use did not change from 2013–2020.

**Table 1.** Land cover classification in Geneva.

| Class | Percentage (%) | Reclassified Groups | Source |
|---|---|---|---|
| Standing water, stream and reed bed | 14.75 | Water | |
| Buildings | 5.07 | | |
| Coated surfaces | 13.43 | Impervious surfaces | SITG * |
| Surfaces without vegetation | 0.82 | | |
| Wooded areas | 15.04 | Green area | |
| Green surfaces | 50.89 | | |

* https://ge.ch/sitg/; Supplementary material: www.bfs.admin.ch [16].

**Table 2.** Land cover classification in Paris.

| Class | Percentage (%) | Reclassified Groups | Source |
|---|---|---|---|
| Water | 2.50 | Water | |
| Housing | 39.02 | | |
| Activities | 6.73 | | |
| Equipment | 12.22 | Impervious surfaces | Institut Paris Region * |
| Transports | 13.71 | | |
| Artificial open spaces | 17.72 | | |
| Quarries, landfills, construction sites | 0.57 | | |
| Wood or forest | 7.34 | | |
| Semi-natural | 0.04 | Green area | Institut Paris Region * |
| Agricultural space | 0.16 | | Paris data ** |

* https://data-iau-idf.opendata.arcgis.com/; ** https://opendata.paris.fr/pages/home/.

### 2.2. Datasets

Land cover data from official statistics and open sources were used as main data source. Satellite images from Landsat 8 were used to derive LST and indices.

#### 2.2.1. Satellite Imagery

Landsat imagery was used to estimate surface biophysical parameters, including LST, NDVI, MNDWI and NDBI. Landsat 8 [18,19] has two sensors: the Operational Land Imager (OLI) and Thermal-Infrared Sensor (TIRS). TIRS, the newest thermal infrared sensor, provides two adjacent thermal bands, e.g., band 10 (10.6–11.19 μm). The resolution is 100 m. OLI uses long linear detector arrays with thousands of detectors per spectral band, collecting image data for 9 shortwave spectral bands. The resolution is 30 m. The characteristics of Landsat imagery used in this study are shown in Tables A1 and A2. The images were filtered according to the land cloud cover and the scene cloud cover. No further atmospheric corrections were performed, as it was assumed that the atmospheric conditions were homogeneous within the study area. A total of 30 satellite images in the period of 2013–2020 were obtained for Geneva and 27 images for Paris. The one-day images were averaged to obtain seasonal scenes.

#### 2.2.2. Land Cover Data

The land cover maps of Paris and Geneva were created (Figure 2).

An overview of land cover data acquired by SITG [20] in Geneva is given in Table 1. Six classes are considered, which were then reclassified into three groups: green area, impervious surfaces and water.

The land cover data of Paris was downloaded from the Institut Paris Region website [18] Ten classes were included in the dataset. The statistics of the land cover and reclassified groups are presented in Table 2. A dataset of green area in Paris was used as supplementary data set, acquired from Paris Data [19].

### 2.3. Land Surface Temperature

The land surface temperature was estimated using the Landsat 8 thermal bands. Remote sensing data with resolution of 30 m from the NASA Landsat-8 satellite were used and band 10 radiance images were extracted. The NDVI based method was used to calculate the proportion of vegetation $P_v$, emissivity $\varepsilon$ and black body radiance. LST was finally obtained by the the Radiative Transfer Equation method [21]. Daily LST images during 2013–2020 were averaged to obtain seasonal LST images.

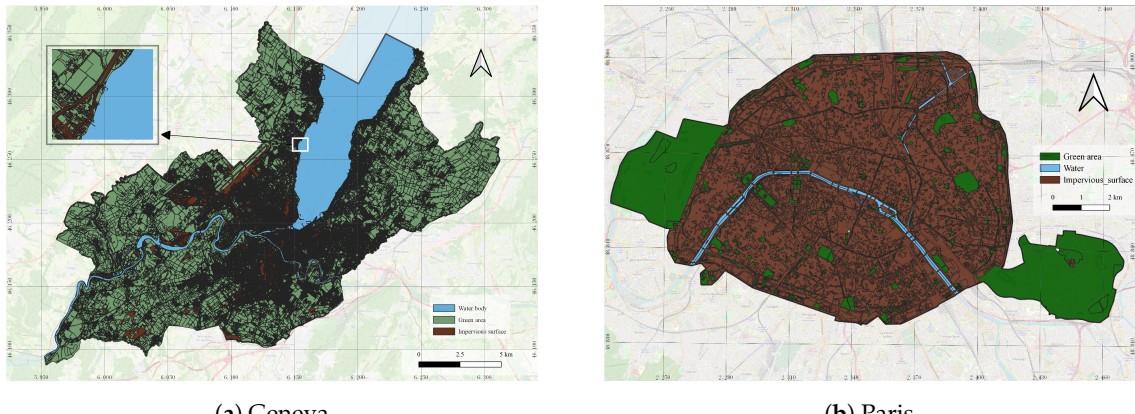

(**a**) Geneva          (**b**) Paris

**Figure 2.** Land cover data.

### 2.4. NDVI, NDBI and MNDWI Computation

Surface biophysical parameters are often considered as driving factors on LST variation [22]. They contain a large amount of spectral information and are easy to obtain from remotely sensed images. Indices of NDBI, NDVI and MNDWI were created to further explore the relationship between LST and land cover features on a smaller scale. NDVI is one of the most widely used indices in detecting vegetation coverage [23]. Here, the bands 4 and 5 were used [24]. The NDBI is a sensitive indicator of built-up or impervious surfaces [25], where bands 5 and 6 were used . The MNDWI [26,27] was used in this study as it improves the accuracy of extracting a waterbody from an urban area [28]. For Landsat 8 images, bands 3 and 6 were used.

### 2.5. Distance-Based Analysis

This method examines the spatial patterns of LST from city center to urban area on a large scale. Four transects (profiles) per city were identified across the image through the center of the city: 0° (from West to East), 45° (from Southwest to Northeast), 90° (from North to South) and 135° (from Northwest to Southeast) in Paris, and 0° (from West to East), 45° (from Southwest to Northeast), 90° (from North to South) and 135° (from Northwest to Southeast) in Geneva. The transects are shown in Figure 3.

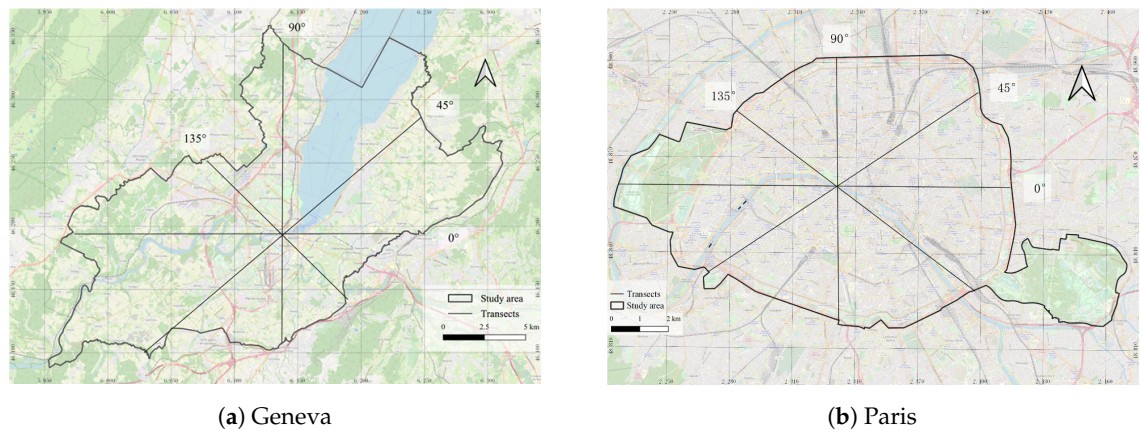

(**a**) Geneva          (**b**) Paris

**Figure 3.** Four transects in Geneva and Paris.

### 2.6. Grid-Based Analysis

To further explore the influence of different land covers on LST on a large scale, 400 m × 400 m polygon grids were created (Figure 4), which is the standard size of a city block. After these sets of grids had been prepared, the mean LST and the density of land cover for each grid were computed using Equation (1). The 400 m × 400 m grid cells were used to create a density map of impervious surfaces and green area.

$$\text{density} = \frac{\text{area of land cover [ha]}}{\text{area of the grid [km}^2\text{]}} \tag{1}$$

For each grid cell, the density of impervious surfaces over the surface of the grid cell [ha/km$^2$] and the density of green area over the surface of the grid cell [ha/km$^2$] were computed, respectively. The analysis examined the correlation between LST and density of impervious surfaces and green area, respectively. The grids outside of the selected study area were not included in this study.

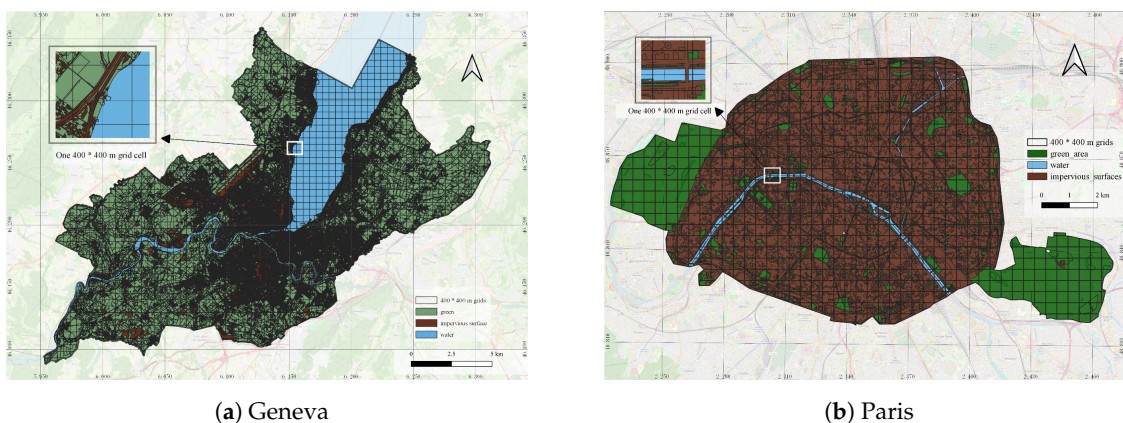

(**a**) Geneva
(**b**) Paris

**Figure 4.** 400 m × 400 m grids.

## 2.7. Point-Based Analysis

In order to explore the spatial correlation between land cover features and LST on a smaller scale, a regressive spatial model was applied. In this part, a total of 10,000 random points were generated across Paris and 25,000 across Geneva. For each point, LST, NDVI, NDBI, and MNDWI were computed at a resolution of 30 m [15]. The Spatial Lag Model was applied, which is a classic spatial regression model, recommended in Qiao's study [28] in comparison to the Spatial Error Model and the Ordinary Least Squares model. The estimated SLM model with exogenous explanatory variables of NDBI, NDVI MNDWI is as follows:

$$LST = \alpha + \rho WLST + \beta_1 NDVI + \beta_2 NDBI + \beta_3 MNDWI + \varepsilon \tag{2}$$

where: $\alpha$ is the model intercept; $\rho$ is the spatial autoregressive coefficient [28]; $W$ is the spatial weight matrix, the method of inverse distance weights was used to create the weight matrix [29]; $\beta_1$, $\beta_2$ and $\beta_3$ are the deviation rate coefficients for NDVI, NDBI and MNDWI, respectively; and $\varepsilon$ is a random disturbance. The SLM mainly considers the spatial autocorrelation of dependent variables, reflecting that the influencing elements of LSTs will act on critical regions by some kind of spatial mechanism.

## 3. Results

### 3.1. Land Surface Temperature

Figures 5 and 6 display the distribution of estimated LST, based on 30 × 30 m grid cells. A similar pattern is observed during all four seasons: areas with higher LSTs tend to be concentrated in the northern part of the river, where the built-up area situated. Low LSTs are mainly concentrated in the western and eastern parts of Paris, where the green area coverage is quite high. The areas holding higher LSTs are located in the city center, extended to the west in spring and winter and the east in summer and autumn.

The proportions of high-LST (above average) area in Paris are 56.51% (20.2 °C in average), 58.13% (27.85 °C in average), 54.01% (18.27 °C in average) and 47.13% (4.93 °C in average) in spring, summer, autumn and winter, respectively. There is a slight downward trend from spring to winter.

Compared to Paris, the mean values of LST in Geneva display a different patterns for each season. Areas with higher mean LST are located within the city center in Spring and Summer, close to the lake. The high LST areas are observed in the southwest of the city in autumn, and on the west side of the city in winter. In general, mean LST is higher in Spring in Geneva while in Paris, the higher mean LST is observed during summer. From Spring to winter, it can be observed that the highest and the minimum LST all decrease. The proportion of high LST (above average) area in Geneva are 34.93% (17.58 °C in average), 33.42% (26.86 °C in average), 40.14% (20.42 °C in average) and 52.68% (5.9 °C in average) in spring, summer, autumn and winter, respectively. There is a slightly upward trend from spring to winter, which is opposite to Paris.

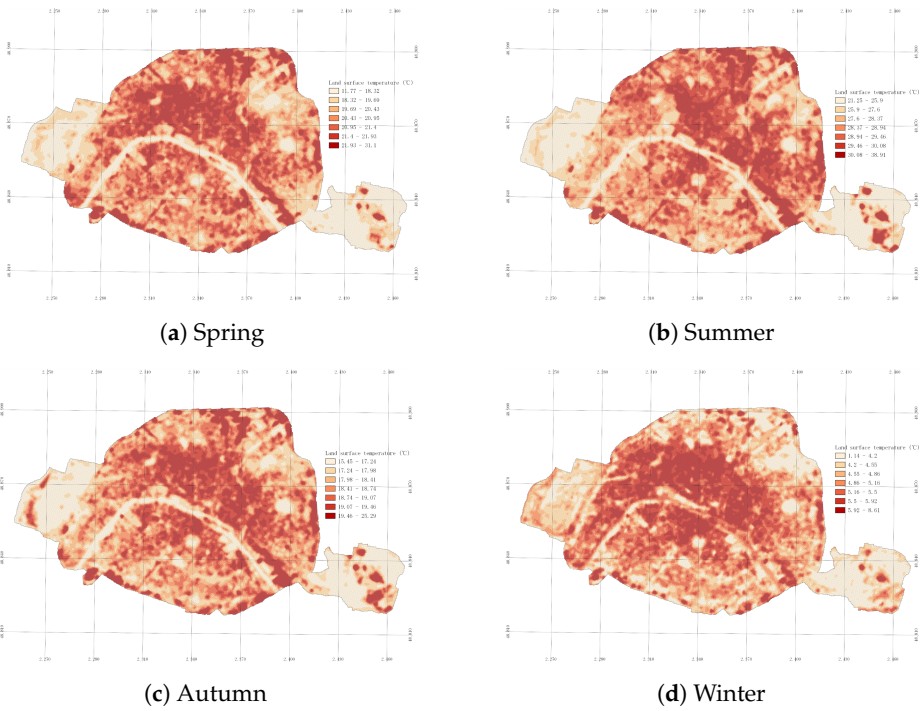

(**a**) Spring　　　　　　　　　　　　　　　　　(**b**) Summer

(**c**) Autumn　　　　　　　　　　　　　　　　　(**d**) Winter

**Figure 5.** Mean Land Surface Temperature (°C) of four seasons in Paris.

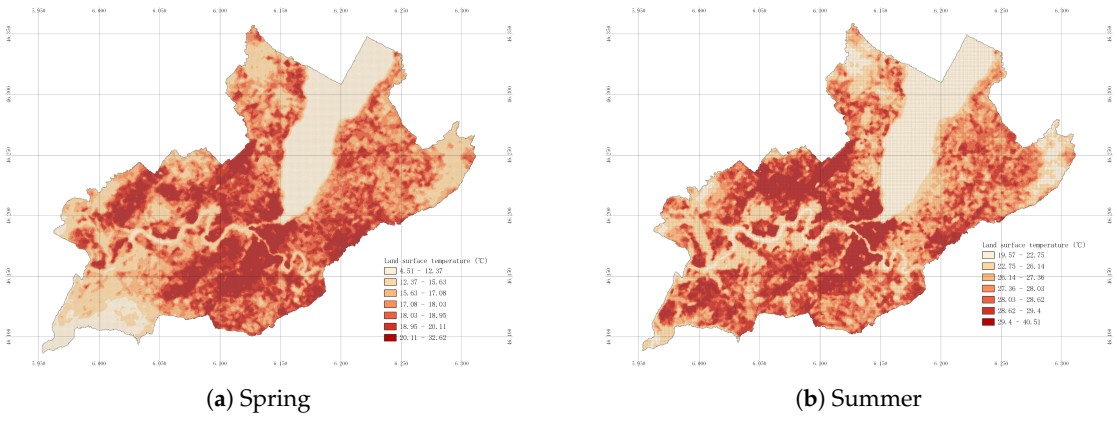

(**a**) Spring　　　　　　　　　　　　　　　　　(**b**) Summer

**Figure 6.** *Cont.*

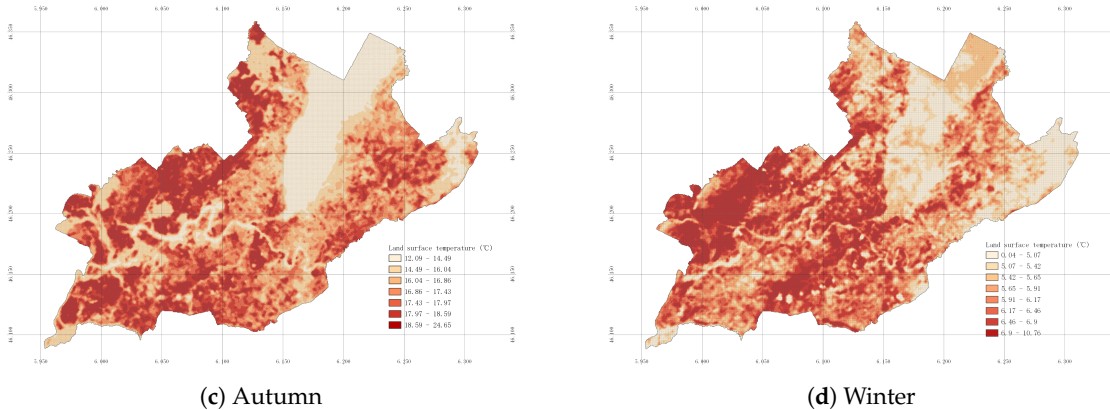

(**c**) Autumn　　　　　　　　　　　　　　　　　　　　　(**d**) Winter

**Figure 6.** Mean Land Surface Temperature (°C) of four seasons in Geneva.

### 3.2. Temporal Distribution of Lst

In the two cities (Figure 7), the widest LST differences among the four seasons are observed for impervious surfaces. The minimum and maximum seasonal LSTs all increase and then decrease from spring to winter. Compared to impervious surfaces and green area, the median LST in the water class and the variation of LST are both observed to be smaller for all the seasons. The largest median LST of all three land cover features is observed in summer. For each season, the largest median LST and the widest variation of LST are observed in the class of impervious surfaces. The opposite pattern is observed in the class of water. There is no significant difference of median LST amongst the three land cover features in Winter both in Paris and Geneva. The statistics are presented in Tables A4 and A3 in Appendix A.

Within the same season, as presented in Figure 8, the median LST and the variation of LST for green area are generally significantly smaller ($p < 0.01$) than those of impervious surfaces both in Paris and Geneva. In addition, wider LST differences between green area and impervious surfaces are observed in Paris (the median LST of green area is around 1.5 °C lower than the impervious surfaces) in comparison to Geneva (the median LST of green area is around 0.5 °C lower than that of impervious surfaces).

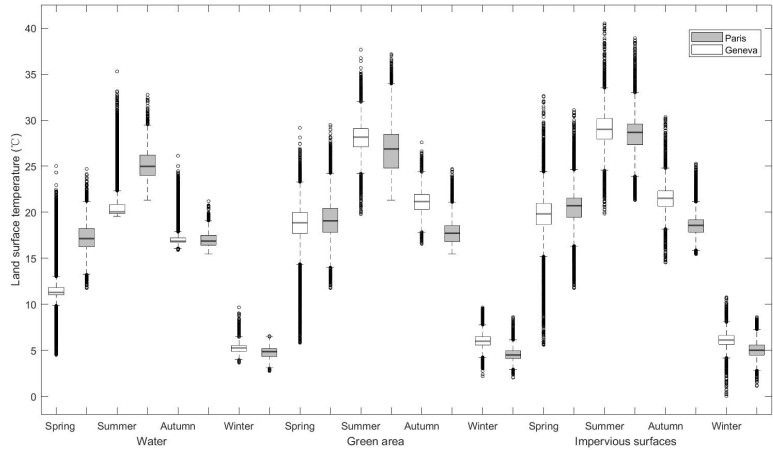

**Figure 7.** Land surface temperature (°C) of three land cover features for each season in Paris and Geneva.

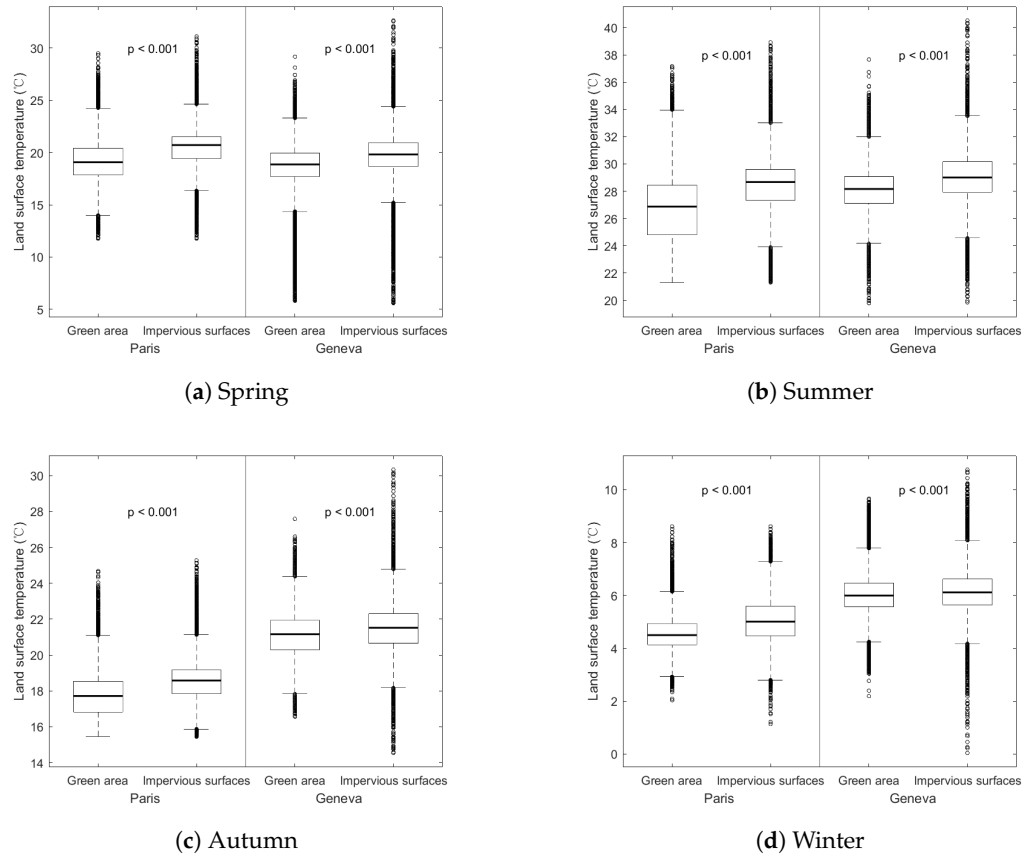

**Figure 8.** Box charts related to land surface temperature of green area and impervious surfaces for four seasons observed in Paris and Geneva.

### 3.3. Distance-Based Results

Similar patterns of LSTs were observed along the four transects with angles of 0, 45, 90 and 135°. Therefore, only the results along the transect of 90° from North to South crossing the city center are presented here. Both in Geneva (Figure 9) and Paris (Figure 10), a symmetric pattern along the transects was observed: LST values shift from low to high and then to low again. Peaks were always observed in the city center. From the city center outward (both to the south and the north), a significant temperature gradient as a result of the UHI effect can be seen. Troughs are always observed around the city center of the two cities, where large waterbodies are situated.

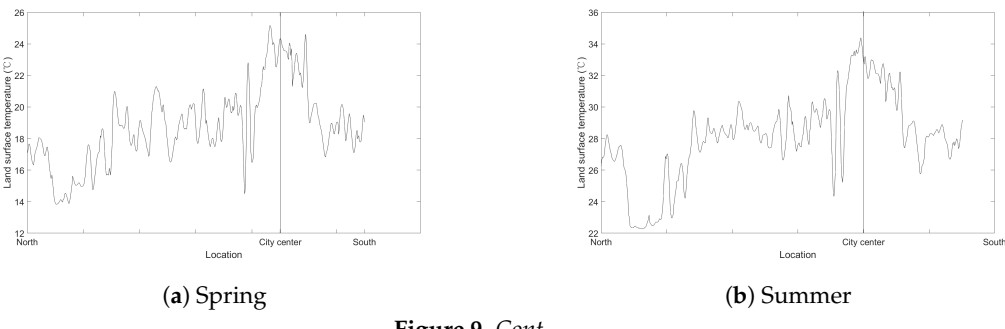

(**a**) Spring                     (**b**) Summer

**Figure 9.** *Cont.*

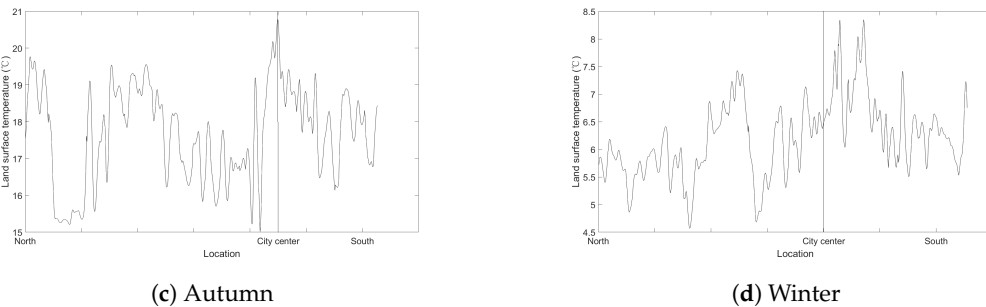

(**c**) Autumn　　　　　　　　　　　　　　　　(**d**) Winter

**Figure 9.** Seasonal land surface temperature (°C) along the north to south transect in Geneva.

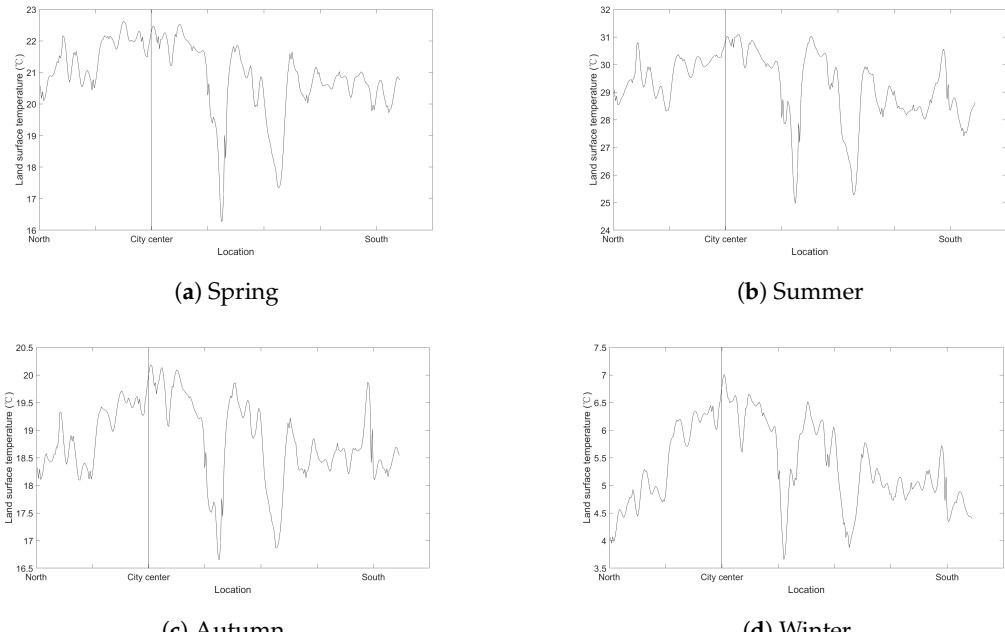

(**a**) Spring　　　　　　　　　　　　　　　　(**b**) Summer

(**c**) Autumn　　　　　　　　　　　　　　　　(**d**) Winter

**Figure 10.** Seasonal land surface temperature (°C) along the north to south transect in Paris.

*3.4. Grid-Based Results*

As shown in Figures 11 and 12, the dark red area represents higher LST. The light red area represents the lower LST. Circles with larger radius represent areas with greater impervious surface density. In Paris, areas with greater impervious surface density are located along River Seine, associated with higher LST. The light red areas are generally covered by smaller radius circles, indicating smaller density of impervious surfaces. The bar chart describes the coverage of each density group. The group with higher density of impervious surfaces has a greater proportion of coverage. The mean LST of each density group increased significantly ($p < 0.01$) along the four seasons. Among the four seasons, the largest difference between the highest mean LST and the lowest mean LST was observed in summer, reaching 2.9 °C, around 0.7 °C per 20 ha/km$^2$ increase of impervious surface density. The smallest difference is observed in winter, equaling 0.8 °C, corresponding to around 0.2 °C per 20 ha/km$^2$ increase of impervious surface density. The largest error bar is observed in the density group of 20–40 ha/km$^2$, with only 1.2 % coverage in the whole study area. More than 70% of the grids are located within the density group of 80–100 ha/km$^2$, showing the smallest error bar.

In Geneva, areas with greater impervious surface density are not always associated with a high LST. The distribution of impervious surfaces is similar to that of Paris. The higher impervious surface densities are located around Lake Geneva, where the pattern of LST changed with season. The city center, connected with Lake Geneva, has a higher impervious surface cover and a relatively higher

LST. A low LST is observed in the southwestern part of Geneva in spring, where around one third of the area showed a higher impervious surface density. The east bank of Lake Geneva, where the LST is lower, showed higher impervious surfaces densities in summer, autumn and winter. The group with a smaller density of impervious surfaces covered a larger proportion, which is opposite of the results for Paris. The mean LST of each density group increases significantly ($p < 0.01$ in spring, summer, $p < 0.05$ in autumn and winter) with the LST for all seasons. The largest difference between the highest mean LST and the lowest mean LST is observed in spring, equally 5.1 °C, which corresponds to around 1.3 °C per 20 ha/km$^2$ increase of impervious surfaces density. The smallest difference is observed in winter, equally 0.6 °C. The temperature differences in spring and summer in Geneva are greater than those found in Paris, while the LST differences in autumn and winter are similar to the ones of Paris.

As shown in Figure 13, most of the areas with greater green area density are situated in the west and east of Paris. Few are located in the northeastern part of the city, where LSTs are very low. Generally, areas with a larger green area density are associated with lower LST values. The largest coverage of green area is observed in the density group of 0–20%, which is near 70% (Figure 13e), while the second largest coverage is observed in the density group of 80–100%. The mean LST of each density group decreases significantly with LST ($p < 0.01$ and $p < 0.05$). Similar to the pattern of impervious surfaces, the largest difference between the highest mean LST and the lowest mean LST is observed in summer, equally 3.3 °C, which corresponds to around 0.8 °C per 20 ha/km$^2$ decrease of green area density. The smallest difference (0.8 °C) is observed in winter. The differences in LST in summer and autumn measured for green area were larger than those for impervious surfaces during the same season. The LST differences in spring and winter were very similar. The largest two error bars are observed in the density group of 40–60 ha/km$^2$ and 60–80 ha/km$^2$, with coverage less than 5% in total.

Contrary to the distribution of impervious surfaces in Geneva, greater green area densities are situated close to the edge of the city (Figure 14) where impervious surface densities are smaller. Generally, most greater densities of green area are correlated with lower LST, except for winter. The western part of Geneva, where LSTs are very high, is almost fully covered by larger densities of green area. The group with a larger density of green area has a larger proportion of low LST (Figure 14e). The mean LST of each density group went up firstly, and then went down in spring and summer. There is no general rule for the pattern of LST in autumn and winter. The overall LST increased by 0.9 °C in autumn, and 0.02 °C in winter. An unusual value, similar to Paris, is also observed in the density group of 40–60 ha/km$^2$ in winter. The largest error bar is observed in the density group of 0–20 ha/km$^2$, with around 10% coverage.

The overall patterns of green area with LST in Geneva are different from Paris. Comparing impervious surfaces and green area maps, the area covered by smaller dots in green area maps (Figure 14a–d) is covered by larger dots as shown in impervious surfaces maps (Figure 12a–d). LST decreases with green area density, while it increases in autumn.

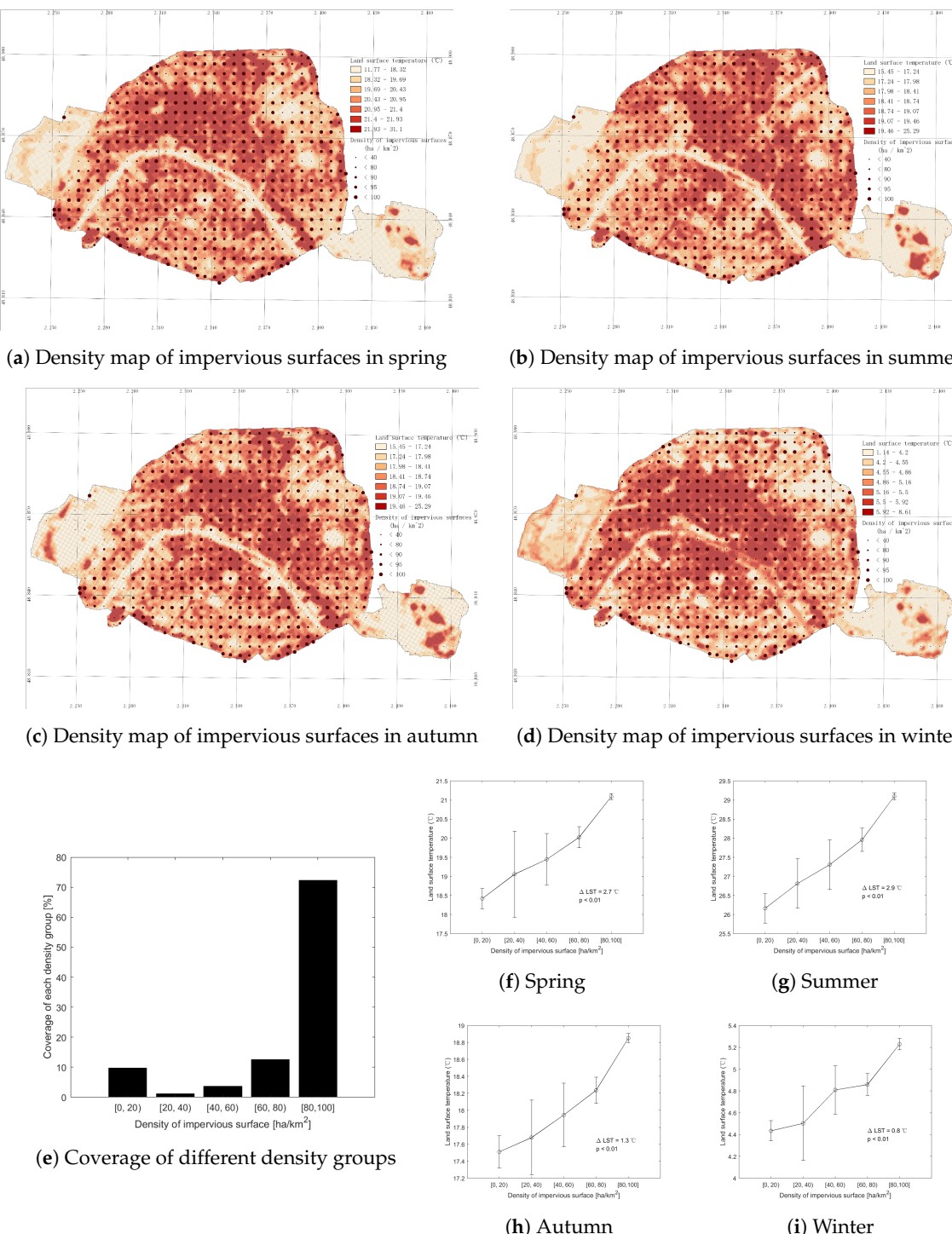

**Figure 11.** Density map of impervious surfaces for four seasons in Paris. Different colored areas and sized circles indicate various temperatures (°C) and different densities respectively. Figures (**f–i**) present the relationships between LST and density of green area with 95% confidence intervals.

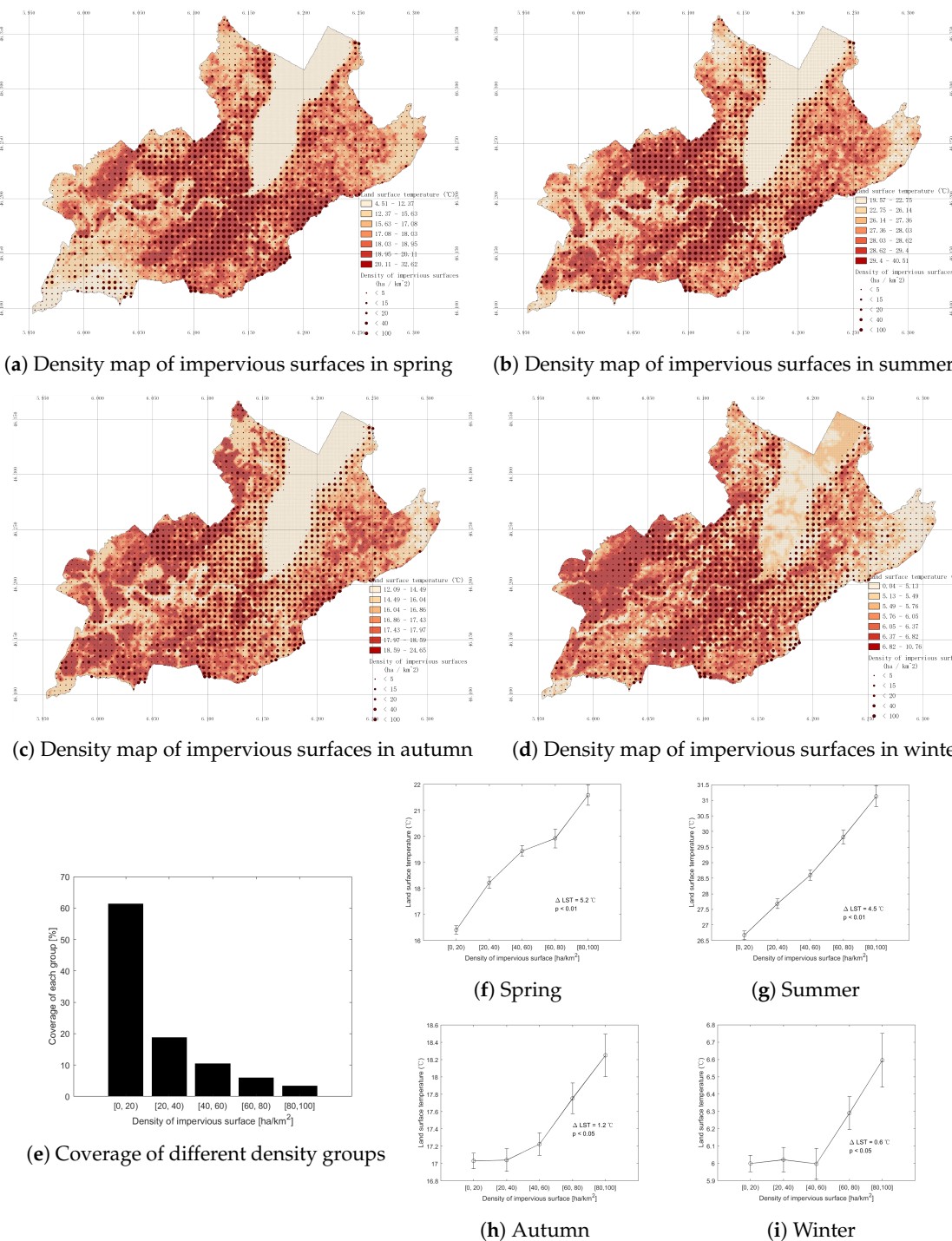

**Figure 12.** Density map of impervious surfaces for four seasons in Geneva. Different colored areas and sized circles indicate various temperatures (°C) and different densities respectively. Figures (**f**–**i**) present the relationships between LST and density of green area with 95% confidence intervals.

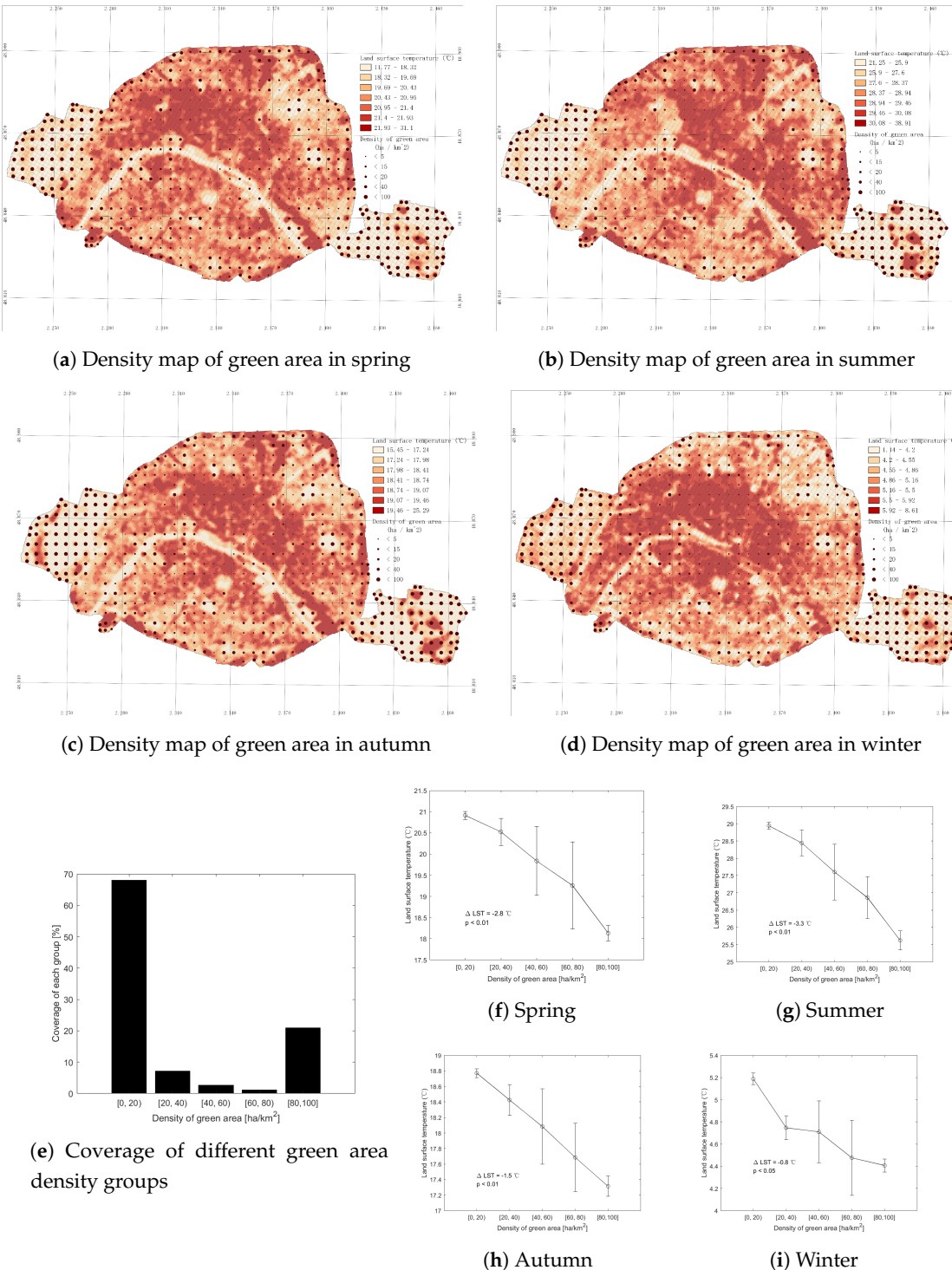

**Figure 13.** Density map of green area for four seasons in Paris. Different colored areas and sized circles indicate various temperatures (°C) and different density groups respectively. Figures (**f–i**) present the relationships between LST and density of green area with 95% confidence intervals.

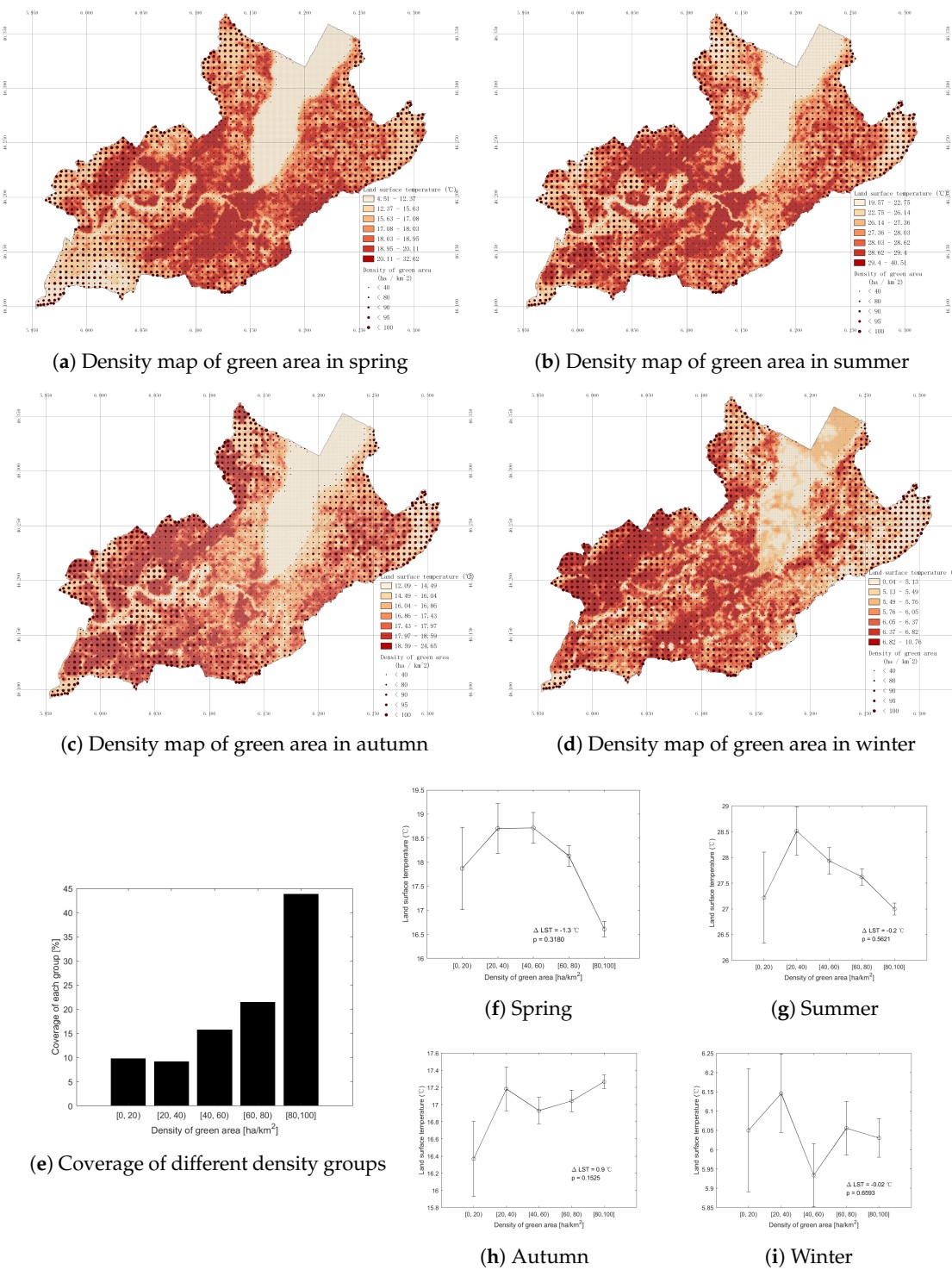

**Figure 14.** Density map of green area for four seasons in Geneva. Different colored areas and sized circles indicate various temperatures (°C) and different density groups respectively. Figures (**f**–**i**) present the relationships between LST and density of green area with 95% confidence intervals.

### 3.5. Point-Based Results

All the indices are negatively correlated with LST at the significance level of 0.01 in both Paris and Geneva, as shown in Table 3. The coefficients of WLST are around 0.9. The results of WLST coefficients proved the positive impact of WLST. The higher the LST in the surrounding area, the higher the LST will be in the core area. The indices have different influences on LST in different seasons. The MNDWI, with the largest absolute value of coefficient in most seasons, always displays the greatest mitigation

on LST. It is followed by NDVI, with the coefficient values slightly smaller than MNDWI. An exception is observed for winter where the effect of NDVI is slightly more significant than MNDWI. The NDBI has the smallest impact on LST. An inverse correlation between NDBI and LST is observed in Paris. The results are different from Geneva, and inconsistent with the aforementioned impervious surface density map of Paris (Figure 11). The pattern of NDBI in Geneva is different from Paris. Significantly positive correlations between NDBI and LST are observed in spring, summer and winter, but negative correlation is observed in autumn.

**Table 3.** Coefficients of indices with LST for Paris

| Variables | Coefficients | | | | | | | |
| | Paris | | | | Geneva | | | |
| | Spring | Summer | Autumn | Winter | Spring | Summer | Autumn | Winter |
|---|---|---|---|---|---|---|---|---|
| WLST | 0.8858 | 0.8949 | 0.8966 | 0.9380 | 0.9205 | 0.9629 | 0.8541 | 0.9440 |
| Constant | 2.5820 | 3.2062 | 2.1065 | 0.3843 | 1.6081 | 0.9629 | 2.1065 | 0.1696 |
| NDBI | −9.5530 | −4.3836 | −4.1051 | −3.4244 | 1.6134 | 2.9515 | −2.5516 | 6.2503 |
| MNDWI | −11.9005 | −8.1810 | −6.5245 | −3.5342 | −5.2034 | −6.2199 | −8.6041 | 3.7929 |
| NDVI | −11.4616 | −7.2408 | −5.8174 | −3.8537 | −2.1090 | −2.5941 | −5.8284 | 5.6238 |
| $R^2$ | 0.8640 | 0.8982 | 0.8425 | 0.8192 | 0.9624 | 0.9629 | 0.9593 | 0.8913 |

All the indices are significantly correlated with LST ($p < 0.01$).

## 4. Discussion

### 4.1. Effects of Waterbody on Land Surface Temperature

From a large to a smaller scale, waterbody density is generally negatively correlated with LST. At a large scale, both in Geneva (Figure 9) and Paris (Figure 10), troughs are always observed around the city center for all seasons, as the city centers are situated along Lake Geneva and the river Seine, respectively. The LST related to waterbody is usually lower than that of other kinds of land cover features [30]. The transect in Geneva crosses grids with different proportions of waterbody. As presented in the Figure 3a, the transect crosses a small size of waterbody in the northern part, leading to a lower point along the LST gradient. The variation of LST in the northern part is smaller than that around the city center, which is affected by the wide surface of the lake. The influence of waterbody size has been explored in previous studies [30]. A negative correlation was observed between the proportion of waterbody and LST, meaning that in a fixed region, the lower the ratio of waterbody, the higher the LST. As shown in the study [31], the average size of the water body, the proportion of the total landscape composed of the largest water area, the isolation and fragmentation of the waterbody are all negatively correlated with the average LST. Except for the influencing factors of the surroundings, the total size and amount of core water has a greater impact on the LST, indicating that the interior waterbody has little external influence. The decrease of LST in Geneva is more significant than the decrease in Paris. The considered area of Lake Geneva is far larger than the River Seine. The decrease of LST around the city center is around 10 °C in Geneva from spring to autumn (2 °C). In Paris, however, the decrease is around 3.5 °C in spring and autumn, 6 °C in summer and 2 °C in winter. On the smaller scale, the MNDWI coefficient is relatively higher in winter than in other seasons. The pattern can be explained by the heat capacity of the waterbody. A water body retains more heat than the surroundings in winter, and therefore maintains a minimum temperature differential.

### 4.2. Effects of Impervious Surface on Land Surface Temperature

Throughout the seasons, the LST increases with the impervious surface density both in Geneva (Figure 12e) and Paris (Figure 11e) on the large scale. The results are consistent with those from previous research where increasing impervious surfaces significantly raises the magnitude of LST, exacerbating the urban heat island (UHI) phenomena [12]. This happens as impervious surfaces have different albedo values and low concentrations of water. The influence of evapotranspiration and

shading of buildings can significantly increase the amount of heat that would be re-radiated by façades and hard surface.

### 4.3. Effects of Green Area on Land Surface Temperature

The results of Paris (Figure 13) showed that green area are negatively correlated with LST and the correlation is statistically significant. This is consistent with numerous previous studies, which identified a negative correlation between LST and street green or vegetation, considering proportion of landscape, edge density and patch density [32], or fraction of vegetation [33]. The main causes for the decrease of LST within green area, compared to impervious surfaces, can be the low heat storage capacity and heat losses through evaporation and transpiration processes in the green area. Green area can lower the LST by providing shade to prevent land surfaces from direct heating during daytime, which was confirmed in Zhou's [12] study. An unusual value is observed in the density group of 40–60 ha/km$^2$ in winter and in the impervious surface density map. It could be a result of the combined effect of green area and impervious surfaces of similar proportions. The results indicate that a small grid mixed with similar proportions of different land cover may regulate the micro-climate, which is easy to observe when the variation of LST is not obvious.

The overall patterns of green area with LST differs between Geneva and Paris and also between this and previous studies [12,32,33]. In Geneva, the temperature significantly fluctuates around its mean value in spring, summer and winter, which can be explained by the patch density and structure and composition of green area. Li's study [34] found that the patch density of green area had a significant positive relationship with LST. An increasing of patch density will increase the LST. The more fragmented the green area, the higher the LST, according to Zhang's study [35]. Liu's study showed that during high temperature periods, the land surface temperature above the green area decreased with increasing coverage, but the significant ecological effects of lowering the temperature were not observed until the coverage of the spaces exceeded 60%. A tree-shrub-herbage mixture with higher coverage had a greater effect on lowering temperature compared to lawn. A 41.3% fraction of Geneva is covered with agricultural land, which is classified as a green area in this case. Land conversion to cropland is not significantly associated with the cooling effect, as found in Mueller's study. In addition, it must be considered that 33.7% of Geneva is composed of woods, with beech as the most common tree species, followed by maple and ash [36]. Most of trees are deciduous. Generally, forest with a multilevel canopy has a low albedo, as the incident radiation can penetrate deeply into the forest canopy where it bounces back and forth between the leaves and branches and gets trapped by the canopy [37]. A positive relationship can be observed between canopy and land surface temperatures[38]. The impact of leaves on the LST was more significant than the impact of trunks. In autumn, leaves fall from the trees and cover the ground, indicating the litter layer [39]. Microbial activity in the litter layer, such as bacterial and fungal dominance in heterotrophic respiration of surface litter [40], may contribute to the increase of temperature. In general, a complicated mixture of green area such as that present in Geneva increases the potential for evapotranspiration and therefore lead to cooling effects. As presented in Figure 7, compared with Paris, the land surface temperature difference is not very large in Geneva.

On the smaller scale, a positive coefficient of NDVI was observed in Geneva's winter (Table 3). The negative correlation between NDVI and LST indicates that the higher the NDVI, the stronger the mitigation of LST. However in winter, green area can reduce exposure to winter cold, thus increasing local land surface [41,42]. Some evergreen vegetation can play a role in the positive impact. The areas with high NDVI (especially above 0.5) are likely to include large amounts of evergreen vegetation, effectively mitigating the effects of cold wind [41].

### 4.4. Combined Effects

The density group of green area between 40 and 60 ha/km$^2$ in winter has the lowest land surface temperature, which is 0.2 °C lower than the highest one. The density group was extracted

and presented in Figure 15. The dots represent green area within the density group. The blue area represents the waterbody distribution for Geneva. The majority of dots are situated around waterbodies. This study [43] found that waterbodies can influence the exchange of energy and modify the transfer of energy and moisture between the land surface and the atmosphere. Surface wetness and moisture play a significant role in reducing the LST, especially in dry seasons [44]. The wetness refers to the amount of soil moisture. Even green area with reduced areas can regulate microclimate, increasing LST and moisture [45]. In winter, the land surface is largely dry and barren in Geneva [46]. Considering the distribution of the density groups, wetness here is an important driving factor in determining LST compared with other density groups. A relatively lower land surface temperature was therefore observed.

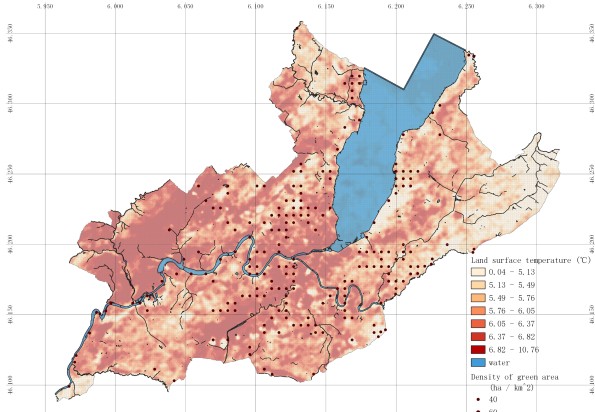

**Figure 15.** Density of green area between 40 and 60 ha/km$^2$ in winter in Geneva.

On the smaller scale, NDVI and MNDWI are always the two most important factors affecting LST among all the influencing factors [22] in all the seasons. An inverse correlation between NDBI and LST was observed for all seasons in Paris and during autumn in Geneva (Table 3). The results are not consistent with the impervious surface density map (Figures 11 and 12). Bare soil in rural areas can play a significant role in this case, which shows high spectral reflectance in the SWIR band compared to urban built-up areas [47]. The pattern can also be observed from drier or sparse vegetation, indicating positive values in NDBI map. The NDBI performance can be adversely affected by the surroundings [48,49], e.g., soil, wetness, size of area, etc. Higher NDBI values would be therefore derived in the rural areas, compared to urban areas, indicating a negative correlation with LST. In addition, around 90 km$^2$ of Geneva is covered with green roofs [20], which could be detected as NDVI, other than NDBI. Green roofs of buildings can effectively lower LST and alleviate UHI effects in high-density urban areas [50], leading to the inverse correlation.

In the case of Geneva, the coefficient of NDBI in winter (6.2503) is larger than that in summer (2.9515). The warming effect is more significant in winter. However, the result differs from other studies [22] where NDBI showed the largest independent contribution rate to LST variation in summer. The dominant impact of a built-up area on the LST gradually declined, with the gradual decrease of LST from summer to winter. The difference of sunlight condition, climate condition and spatial characteristics of built-up areas of different cities can lead to the uncertainty of this study on both LST spatial variation and dominant driving factors.

## 5. Conclusions

In this study, the seasonal LST variation and the correlation between spatial variability of LST and land cover features (i.e., impervious surface, green area and waterbody) were investigated using a distance-based analysis, a grid-based analysis and a point-based analysis. Two scales have been explored: large (400 m × 400 m) and smaller (30 m × 30 m). Both in Paris and Geneva, the results confirm that the highest LST was observed in the city center for all seasons. The dominant driving

factor of LST variation is the presence of waterbody. With different spatial distributions of impervious surface, a significantly positive correlation was observed between the mean LST and the impervious surface density in the two cities. The inverse correlation between mean LST and green area density was observed in Paris for all seasons. It was not possible to draw full conclusions with regard to the correlation of the mean LST and green area density in Geneva, as a city with a complex mixture of land cover features. Generally, the mean LST decreases with the increase of green area density in spring, summer and winter. SLM was used to explore the spatial correlation between LST and three indices (i.e., NDBI, NDVI and MNDWI) on the smaller scale. Inverse correlations between LST and NDVI and MNDWI, respectively, were observed from spring to winter both in Paris and Geneva. There is no obvious correlation between NDBI and LST on the smaller scale, due to the fact that they are affected by an ensemble of factors, e.g., climate conditions, topography and land cover classification. Indeed, the spatial variation of LST is influenced by combined effects of various driving factors [51,52]. Generally, a waterbody is a dominant driving factor on lowering LST and mitigating UHI effects on the large and smaller scale. Green area plays an important role in cooling effects on a smaller scale both in Paris and Geneva.

Land use change (growth of urban agglomerations, transformation of wetland into artificial land, and decrease of agricultural land) in the study areas contributes to the increase of uncertainty in the analysis, especially on the smaller scale. It must be noticed, however, that while locally these modifications can be significant, there are no noticeable changes at the scale of the city. Additionally, the estimation of LST from thermal images can be more complex than the method used in this study. The images were acquired with a land cloud cover and scene cloud cover of less than 10%, respectively. The effects of the atmosphere and surface roughness on the LST were therefore not taken into account. Although atmospheric correction (which may introduce an error of 4–7 °C for the mid-latitude summer atmosphere [53]) was not performed here, we are confident that the selection of the data with little or no cloud cover limited the impact of the correction. Empirical NDVI values of soil and water, and shape factor were used to estimate the proportion of vegetation and emissivity. For example, snow usually shows very low NDVI values (e.g., 0.1 or less), which is lower than the empirical value of 0.2 used in this study, contributing to the uncertainty of the estimated LST and the interpretation of LST patterns [24]. In addition, different climatic conditions may significantly influence the amplitude of daytime LST [12]. Limited by data acquisition for some seasons, the study should be further extended to more deeply investigate the influence of built-up areas on the LST. Based on the similar behavior observed for Paris and Geneva, we can nevertheless conclude that future urban planning scenarios should include an increase of evenly distributed green area and waterbodies at the scale of the city, to lower LST and mitigate UHI effects.

**Author Contributions:** Conceptualization, Xu Ge and Dasaraden Mauree; methodology, Xu Ge and Dasaraden Mauree; software, Xu Ge; validation, Xu Ge, Dasaraden Mauree and Roberto Castello; formal analysis, Xu Ge; investigation, Xu Ge; resources, Xu Ge, Dasaraden Mauree and Roberto Castello; data curation, Xu Ge; writing—original draft preparation, Xu Ge; writing—review and editing, Xu Ge, Dasaraden Mauree and Roberto Castello; visualization, Xu Ge; supervision, Dasaraden Mauree, Roberto Castello and Jean-Louis Scartezzini; project administration, Dasaraden Mauree, Roberto Castello and Jean-Louis Scartezzini; funding acquisition, Dasaraden Mauree, Roberto Castello and Jean-Louis Scartezzini All authors have read and agreed to the published version of the manuscript.

**Funding:** This research project has been financially supported by the HyEnergy project of the National Research Programme 75 "Big Data" (PNR75) of the Swiss National Science Foundation (SNSF) and by the Swiss Innovation Agency Innosuisse and is part of the Swiss Competence Center for Energy Research SCCER FEEB&D.

**Conflicts of Interest:** The authors declare no conflict of interest. The funders had no role in the design of the study; in the collection, analyses, or interpretation of data; in the writing of the manuscript, or in the decision to publish the results.

## Abbreviations

The following abbreviations are used in this manuscript:

LST        Land surface temperature
NDBI       the Normalized Difference Built-up Index
NDVI       the Normalized Difference Vegetation Index
MNDWI    the Modified Normalized Difference Water Index
UHI        Urban heat island effects
SLM       Spatial lag model

## Appendix A. Tables

**Table A1.** The characteristics of Landsat imagery for Geneva.

| Seasons | Acquisition Date | Collection Category | Day/Night Indicator | Image Quality | Land Cloud Cover | Scene Could Cover |
|---------|------------------|---------------------|---------------------|---------------|------------------|-------------------|
| Spring | 2018/5/25 | T1 | Day | 9 | 9.39 | 9.39 |
|        | 2018/3/22 | T1 | Day | 9 | 4.36 | 4.36 |
|        | 2017/5/22 | T1 | Day | 9 | 9.9  | 9.9  |
|        | 2017/4/20 | T1 | Day | 9 | 4.43 | 4.43 |
|        | 2013/5/27 | T1 | Day | 9 | 7.88 | 7.88 |
|        | 2013/4/25 | T1 | Day | 9 | 2.97 | 2.97 |
| Summer | 2019/6/29 | T1 | Day | 9 | 1.71 | 1.71 |
|        | 2019/6/13 | T1 | Day | 9 | 3.19 | 3.19 |
|        | 2018/6/26 | T1 | Day | 9 | 4.43 | 4.43 |
|        | 2017/8/26 | T1 | Day | 9 | 7.04 | 7.04 |
|        | 2016/8/23 | T1 | Day | 9 | 0.44 | 0.44 |
|        | 2016/8/7  | T1 | Day | 9 | 3.04 | 3.04 |
|        | 2015/8/21 | T1 | Day | 9 | 2.47 | 2.47 |
|        | 2015/8/5  | T1 | Day | 9 | 0.75 | 0.75 |
|        | 2015/7/20 | T1 | Day | 9 | 6.57 | 6.57 |
|        | 2015/7/4  | T1 | Day | 9 | 1.77 | 1.77 |
|        | 2014/7/17 | T1 | Day | 9 | 1.53 | 1.53 |
|        | 2013/8/31 | T1 | Day | 9 | 8.76 | 8.76 |
|        | 2013/8/15 | T1 | Day | 9 | 1.01 | 1.01 |
|        | 2013/7/14 | T1 | Day | 9 | 3.26 | 3.26 |
| Autumn | 2019/10/3 | T1 | Day | 9 | 3.24 | 3.24 |
|        | 2019/9/17 | T1 | Day | 9 | 3.81 | 3.81 |
|        | 2017/10/13| T1 | Day | 9 | 6.66 | 6.66 |
|        | 2016/9/24 | T1 | Day | 9 | 6.19 | 6.19 |
|        | 2016/9/8  | T1 | Day | 9 | 2.17 | 2.17 |
|        | 2014/11/22| T1 | Day | 9 | 9.95 | 9.95 |
| Winter | 2020/2/24 | T1 | Day | 9 | 7.88 | 7.88 |
|        | 2019/2/21 | T1 | Day | 9 | 1.86 | 1.86 |
|        | 2019/2/5  | T1 | Day | 9 | 3.58 | 3.58 |
|        | 2019/1/4  | T1 | Day | 9 | 7.02 | 7.02 |

**Table A2.** The characteristics of Landsat imagery for Paris.

| Seasons | Acquisition Date | Collection Category | Day/Night Indicator | Image Quality | Land Cloud Cover | Scene Could Cover |
|---------|-----------------|---------------------|---------------------|---------------|------------------|-------------------|
| Spring | 2020/5/19 | T1 | Day | 9 | 0.27 | 0.27 |
| | 2020/4/1 | T1 | Day | 9 | 0.01 | 0.01 |
| | 2019/3/30 | T1 | Day | 9 | 9.42 | 9.42 |
| | 2017/4/9 | T1 | Day | 9 | 0.01 | 0.01 |
| | 2015/4/20 | T1 | Day | 9 | 5.99 | 5.99 |
| | 2014/5/19 | T1 | Day | 9 | 0.1 | 0.1 |
| | 2014/4/17 | T1 | Day | 9 | 5.66 | 5.66 |
| | 2014/3/16 | T1 | Day | 9 | 8.18 | 8.18 |
| Summer | 2019/7/4 | T1 | Day | 9 | 0 | 0 |
| | 2019/6/2 | T1 | Day | 9 | 0.13 | 0.13 |
| | 2018/8/2 | T1 | Day | 9 | 2.82 | 2.82 |
| | 2015/6/7 | T1 | Day | 9 | 9.58 | 9.58 |
| | 2013/8/20 | T1 | Day | 9 | 4.99 | 4.99 |
| | 2013/7/19 | T1 | Day | 9 | 9.59 | 9.59 |
| Autumn | 2019/9/6 | T1 | Day | 9 | 7.73 | 7.73 |
| | 2018/10/21 | T1 | Day | 9 | 0.06 | 0.06 |
| | 2018/10/5 | T1 | Day | 9 | 0.04 | 0.04 |
| | 2016/10/31 | T1 | Day | 9 | 0.03 | 0.03 |
| | 2015/9/27 | T1 | Day | 9 | 8.76 | 8.76 |
| | 2014/11/11 | T1 | Day | 9 | 4.55 | 4.55 |
| | 2014/9/8 | T1 | Day | 9 | 2.36 | 2.36 |
| | 2014/9/8 | T1 | Day | 9 | 2.36 | 2.36 |
| | 2013/9/5 | T1 | Day | 9 | 0.01 | 0.01 |
| Winter | 2019/2/26 | T1 | Day | 9 | 0.04 | 0.04 |
| | 2018/2/23 | T1 | Day | 9 | 0.15 | 0.15 |
| | 2017/1/19 | T1 | Day | 9 | 3.42 | 3.42 |
| | 2013/12/10 | T1 | Day | 9 | 8.95 | 8.95 |

**Table A3.** Statistics of box plot in Paris.

| Statistics | | Lower Outlier | Q1 | Median | Q3 | Upper Outlier | IQR | SD |
|------------|--------|---------------|------|--------|-------|---------------|------|------|
| Water | Spring | 13.25 | 16.23 | 17.14 | 18.22 | 21.20 | 1.99 | 1.64 |
| | Summer | 20.68 | 24.00 | 24.99 | 26.22 | 29.54 | 2.22 | 1.71 |
| | Autumn | 14.83 | 16.43 | 16.87 | 17.49 | 19.09 | 1.07 | 0.82 |
| | Winter | 3.04 | 4.32 | 4.86 | 5.18 | 6.47 | 0.86 | 0.60 |
| Green area | Spring | 13.99 | 17.84 | 19.07 | 20.41 | 24.27 | 2.57 | 1.79 |
| | Summer | 19.30 | 24.79 | 26.88 | 28.46 | 33.96 | 3.67 | 2.31 |
| | Autumn | 14.25 | 16.83 | 17.72 | 18.54 | 21.11 | 1.72 | 1.13 |
| | Winter | 2.93 | 4.13 | 4.50 | 4.94 | 6.15 | 0.81 | 0.63 |
| Impervious surface | Spring | 16.34 | 19.45 | 20.72 | 21.52 | 24.63 | 2.07 | 1.72 |
| | Summer | 23.92 | 27.33 | 28.67 | 29.60 | 33.01 | 2.27 | 2.05 |
| | Autumn | 15.87 | 17.85 | 18.58 | 19.16 | 21.14 | 1.32 | 1.06 |
| | Winter | 2.80 | 4.48 | 5.01 | 5.60 | 7.28 | 1.12 | 0.76 |

Q1, First quartile (25th percentile); Q3, third quartile (75th percentile); IQR, interquartile range; SD, standard deviation.

**Table A4.** Statistics of box plot for Geneva.

| Satistics | | Lower Outlier | Q1 | Median | Q3 | Upper Outlier | IQR | SD |
|---|---|---|---|---|---|---|---|---|
| Water | Spring | 9.84 | 11.04 | 11.29 | 11.84 | 13.03 | 0.80 | 1.86 |
| | Summer | 18.35 | 19.85 | 20.05 | 20.85 | 22.35 | 1.00 | 1.94 |
| | Autumn | 16.09 | 16.76 | 16.86 | 17.21 | 17.88 | 0.45 | 1.06 |
| | Winter | 4.00 | 4.92 | 5.26 | 5.54 | 6.46 | 0.62 | 0.55 |
| Green area | Spring | 14.36 | 17.72 | 18.86 | 19.95 | 23.30 | 2.24 | 2.05 |
| | Summer | 24.19 | 27.12 | 28.16 | 29.07 | 32.00 | 1.95 | 1.56 |
| | Autumn | 17.85 | 20.30 | 21.16 | 21.93 | 24.39 | 1.64 | 1.23 |
| | Winter | 4.25 | 5.57 | 5.99 | 6.46 | 7.79 | 0.89 | 0.72 |
| Impervious surface | Spring | 15.20 | 18.65 | 19.82 | 20.95 | 24.41 | 2.30 | 2.11 |
| | Summer | 24.57 | 27.93 | 29.01 | 30.17 | 33.53 | 2.24 | 1.85 |
| | Autumn | 18.18 | 20.66 | 21.52 | 22.32 | 24.80 | 1.65 | 1.32 |
| | Winter | 4.18 | 5.65 | 6.12 | 6.62 | 8.08 | 0.98 | 0.76 |

Q1, First quartile (25th percentile); Q3, third quartile (75th percentile); IQR, interquartile range; SD, standard deviation.

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
