# Peer review of "Spatio-Temporal Relationship between Land Cover and Land Surface Temperature in Urban Areas: A Case Study in Geneva and Paris"

_ijgi, doi:10.3390/ijgi9100593_

Round 1

Reviewer 1 Report

This study is considered to be a meaningful study in that it aimed to analyze the surface temperature characteristics according to the land use type by using satellite images for reducing the surface temperature. But the following modifications are required.

  1. Delete 1.1 context
  2. In the text, fill out the reference in the form.
  3. There are many previous studies that analyzed the relationship between time-series changes in surface temperature and land cover using satellite images. What is the difference from those studies? It is not clear whether the use of point-based, grid-based, and SLM is different from this study. Please clearly show the differentiation of this study.
  4. Overall, the resolution of the picture is low, so the results of the text cannot be confirmed. Please modify the picture in the text.
  5. Line 179~181: It is not point-based analysis, but why did you analyze 100 grid-based?
  6. Tables 1 and 2 are the land use analysis results. It should be included in Section 3.3.2. And overall, it is necessary to modify the order of writing.
  7. Line 227: Why did you resampling?
  8. What is the difference between land cloud cover and scene cloud cover in Tables 3 and 4? The values are the same.
  9. The surface temperature of agricultural areas may be different from the types of vegetation, such as forests and grassland. I think it is not appropriate to classify agricultural areas as green areas.
  10. Line 249: Correct in English.
  11. Figure 11, 12, table 7, 8: In summer, the temperature difference between vegetation and impervious areas is not large. This needs to be compared with the results of other prior studies. This may be why agricultural areas are classified as green areas and the surface temperature is high. In particular, geneva is likely to have a large impact because the agricultural area is 41.3%.
  12. 5 Why did you analyze only 90 degrees in distance-based? Results should be presented in all directions.
  13. Figure 17: Why does impervious density vary from season to season? Does land-use change seasonally?
  14. Figure 20: Why is the 40-60 range low in winter? The patch size and edge described in this study are not well understood.
  15. Table 9, table 10: NDVI and NDBI reflect the surface covering properties. In that sense, why did you analyze point-based and grid-based separately? The results are also different. So which results are more meaningful?

Reviewer 2 Report

This paper analyzes the relationship between land cover and land surface temperature from multiple angles, which is of certain research value, and has the following problems:

1, The introduction section does not elaborate researches on the relationship between LST and land cover.

2, It is recommended that section 3 be placed before section 2.

3, The author should explain why Geneva and Paris are selected as the research areas.

4, What is the relationship between Table1,2,5, and 6? Is it reasonable to assume no change in land use in Paris from 2013 to 2020?

5, In Table 3 and Table 4, why is the amount of data in each season inconsistent? Summer is obviously more than winter.

6, The description in Section 2.5 is inaccurate.

7, The figures are not clear enough, the legend in the figure can not see clearly.

8, The logic of section4 is chaotic, and the analysis section is repetitive and lengthy.

Reviewer 3 Report

Comments for IJGI-893324

It is really a long paper. I don’t think this manuscript is well prepared. It is more like a technical report than article. This manuscript wants to examine the effects of the impervious surfaces and green infrastructure on LST, the distance-based analysis, grid-based analysis and point-based analysis were conducted. The structure of this manuscript needs to be changed. Your method and result must focus on Your aim. What is the different between you study and previous study? Did you find some mew points?

In introduction, you have”1.1. Context”, where is “1.2.”? You downloaded the land cover dataset from some website, I don’t understand what is the reason that you descried the land cover classification.

In Methodology, if you didn’t improve the methods of land surface temperature, you should summaries it and cite the reference due to this method is general for remote sensing. I like to write study area and methodology into the same section.

In Result, please summaries them. Don’t write all the things, the important, and interesting points are enough. The sub title “4.5 Distance-based analysis”, “4.6 Grid-based analysis” and “4.7 Point-based analysis ” are not suitable, they are your method.

In Conclusion, something should write in method, such as “Seasonal LST was averaged based on Landsat 8 satellite images from 2013 to 2020. the remotely sensed data from 2013 to 2020 in Paris and Geneva were divided into Spring, Summer, Autumn and Winter.”

1.The two cities, Geneva and Paris were selected, but why you chosen these two cities?

2.In abstract, line 8, what is the similar pattern? The one key of your manuscript is the spatial characteristic of LST. the result in abstract should be some important points.

  1. Line 92, this is an article not a report.

  1. For different Landsat data, the bands were different for NDVI and NDBI and other indices.

  1. the reason for using 400*400 grids size. And you also use 30*30, are these analysis in the same grids size? If not, what about the scale dependence?

  1. About the land use data of Geneva, at table 1 “2013-2018”, but at conclusion, you write 2020, why?

  1. Page 10, In table 3, What is the difference between Land cloud cover and Scene cloud cover? And as we all known, the satellite has a 16-day repeat cycle with an equatorial crossing time: 10:00 a.m. +/- 15 minutes, thus we only can obtain the day data, is “Day/Night indicator” necessary?

8.Page 12, in table 5, the landcover data sets were download from different sources, what about the consistency of two sources?

  1. You got so many time images, but your landcover data is 2017(2020) for Paris (Geneva), did all images have been used? I can see four season analysis, but you have different years.

10.Line248-250, please write the six class into English.

  1. The digital number of all the tables should be checked to make sure they are in the same pattern.

  1. Check your figures, make sure they are clear.

  1. Page 16, figure 16, please add a line in city center.

  1. did you do further analysis about the relationship between proportion of water/impervious surface/green and LST along your four transects in 4.5? because you conclude these relationships in your result.

15.Figure 17, where is the figure (j)?

16 Page 18, in table 7, what is the meaning of Q1, Q3, IQR and SD? Table 7 and Figure 11 show the same thing? As well as Table 8 and Figure 12? For comparation, combine figure 11 and figure 12 into one figure will be better.

Please check the spelling.

Reviewer 4 Report

Thanks for putting the term "Case Study" in the title and it is a well written paper. However the paper needs several improvements.

Please improve the quality of your images without which this paper should not be published. For many of the images (for example figure 6,7, etc.) the legend is not even readable. 

It is a comparative case study and uses many established indices and methodologies with recent imagery so try to reduce the number of overall pages of the paper. It seems like a very long paper and includes much of the already published content that can be cited. You can also summarize and reduce the results and discussion portion. .

Deriving conclusion such as "the 14.14% decrease of barren land .. introduce a 9 degree C increase of LST," is a huge claim. There are many other factors responsible for increase in temperature so it is wrong to arrive in a conclusion that land use change is the only reason behind the temperature difference. 

Round 2

Reviewer 1 Report

It seems that the amendments have been properly reflected. Please check the typos once again.

Reviewer 2 Report

The author has basically completed the modification of the questions raised, but there are still several minor problems as follows:

1.The abstract does not highlight the key points of the article.

2.There are some grammatical errors in the article after the modification

Reviewer 3 Report

Comments for IJGI-893324

The current version is uneasy to read. Maybe you can accept all changes and highlights the mainly part that have been changed. Your aim is to investigate the influence of impervious surface, green area and water bodies on LST, but for the large scale (400m grids), you only conducted the analysis of the density of impervious surfaces and green areas, where is the water? Please, specify your large scale and smaller scale in the abstract. And, I still think the distance-based analysis is less related with your aim. The high LST were observed in city center due to high density of impervious surface, right? Focus on impervious surface, green area, and water, it will be better.

  1. In abstract: “Generally, water body is a dominant driving factors on lowering LST and…”  but line 602, “The MNDWI, with the largest absolute value of coefficient in most seasons, always displayed the greatest mitigation on LST “, this is your result. In your paper, write your result, not the common sense.
  2. It is inconsistent with your analysis, you write in line 83-86: “(4) to apply the spatial Lag Model (SLM) to investigate the spatial correlation of LST and dependent variables, i.e., NDBI, NDVI, MNDWI, density of impervious surface, density of green area and density of water body.”
  3. Green space and green area, please keep one.

  1. Line 70, CO2, please change it.
  2. In table 2 and table 3, the class names are different, why? They are different dataset?

  1. Lines 339-340, “The grids that do not contain land cover were not included in the analysis”, If impervious surfaces or green areas did not contained in the grid, that grid would be excluded in the analysis? So, the sentences need to be changed.

  1. Line 393, “The NDBI is a driving factor for LST variation”, where can we get it?

  1. Lines 397, The proportion of high LST (average)…… where are the average values?

  1. Lines 406-407: “For each season, the larger median LST and the wider variation of LST are observed in the class of impervious surfaces.” What is the variation of LST, is it the IQR or SD in Table A4 and A3 in Appendix? For Geneva, the wider variations of LST are observed in the impervious surfaces in four seasons, but for Paris, it is green areas, excluding in the winter, referring to IQR and SD. Please make sure that. Or you think the length of boxplot as the variation?

  1. Figure 5. About Mean LST, I know years data were average to obtain the mean value of season, but you still didn’t tell us in section 2.

  1. Lines 831-832: There is no obvious correlation between NDBI and LST. Which result indicates this?

  1. you compared the coefficients of WLST, NDVI, NDBI, and MNDVI, whether they are in the same data range? If the data range is different, the indexes need to be standardized.

  1. About maps, all the maps we plotted have latitude and longitude grids. I am not sure whether the journal have this standard?

  1. Check the reference format.

Round 3

Reviewer 3 Report

Table 1 and Table 3,the class names are different, if you want to use the percentage of each classes, you have the data, you can compute it. Thus, the class names are consistent, and table 1 and table 2 keep the same format. (class name , percentage , reclassify,and source)

Author Response

Thank you for the comment. We have modified the tables accordingly.